# Multi-Level Analysis of Learning Management Systems' User Acceptance Exemplified in Two System Case Studies †

**Parisa Shayan** [1,*], **Roberto Rondinelli** [2], **Menno van Zaanen** [3] **and Martin Atzmueller** [4,5]

1. School of Humanities and Digital Sciences, Cognitive Science and Artificial Intelligence, Tilburg University, 5037 AB Tilburg, The Netherlands
2. Department of Economics and Statistics, University of Naples Federico II, Via Cintia 26, 80126 Naples, Italy
3. South African Centre for Digital Language Resources, North-West University, Potchefstroom 2520, South Africa
4. Semantic Information Systems Group, Osnabrück University, 49090 Osnabrück, Germany
5. German Research Center for Artificial Intelligence (DKFI), 49090 Osnabrück, Germany
* Correspondence: p.shayan@tilburguniversity.edu
† This paper is an extended version of our paper published in ABIS'19: Proceedings of the 23rd International Workshop on Personalization and Recommendation on the Web and Beyond; ACM: Boston, MA, USA, 2019; pp. 7–13.

**Abstract:** There has recently been an increasing interest in Learning Management Systems (LMSs). It is currently unclear, however, exactly how these systems are perceived by their users. This article analyzes data on user acceptance for two LMSs (Blackboard and Canvas). The respective data are collected using a questionnaire modeled after the Technology Acceptance Model (TAM); it relates several variables that influence system acceptability, allowing for a detailed analysis of the system acceptance. We present analyses at two levels of the questionnaire data: questions and constructs (taken from TAM) as well as on different analysis levels using targeted methods. First, we investigate the differences between the above LMSs using statistical tests ($t$-test). Second, we provide results at the question level using descriptive indices, such as the mean and the Gini heterogeneity index, and apply methods for ordinal data using the Cumulative Link Mixed Model (CLMM). Next, we apply the same approach at the TAM construct level plus descriptive network analysis (degree centrality and bipartite motifs) to explore the variability of users' answers and the degree of users' satisfaction considering the extracted patterns. In the context of TAM, the statistical model is able to analyze LMS acceptance on the question level. As we are also very much interested in identifying LMS acceptance at the construct level, in this article, we provide both statistical analysis as well as network analysis to explore the connection between questionnaire data and relational data. A network analysis approach is particularly useful when analyzing LMS acceptance on the construct level, as this can take the structure of the users' answers across questions per construct into account. Taken together, these results suggest a higher rate of user acceptance among Canvas users compared to Blackboard both for the question and construct level. Likewise, the descriptive network modeling for Canvas indicates a slightly higher concordance between Canvas users than Blackboard at the construct level.

**Keywords:** Learning Management System; Technology Acceptance Model; Cumulative Link Mixed Model; descriptive network analysis

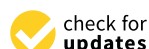



## 1. Introduction

Within the context of higher education, Learning Management Systems (LMSs) are often used to support learning processes. LMSs are software frameworks that provide functionality that helps to share information from the instructor to users, e.g., via catalogs, instructional content, such as learning objectives, assignments, lecture slides, or other course content. Additionally, the system can collect information on the behavior of users and their interaction with the system. This includes, for example, managing registered

users' logins, observing interactions with the provided course material, and in general monitoring users' activities with the system [1].

The effectiveness of the use of LMSs depends heavily on whether users (and instructors) are willing to use the system. However, not everybody may accept the use of LMSs on the same level. In this article, a significantly adapted and extended revision of [2], we specifically aim to investigate how users (in this case, students) perceive and accept the use of two LMSs: Blackboard and Canvas. These LMSs are used at Tilburg University (Tilburg, The Netherlands), where recently, the transition from Blackboard to Canvas took place. With respect to [2], where only Blackboard was analyzed, the research in this article provides a more extensive comparison and an in-depth understanding of the user acceptance of the two systems: Blackboard versus Canvas.

We apply the Technology Acceptance Model (TAM), which was introduced by [3]. TAM is an information systems theory that is commonly used to model users' acceptance of (novel) technologies. The model is adapted from the Theory of Reasoned Action (TRA) [4], which is specifically designed to model user acceptance of information systems [5]. TAM (see Figure 1) consists of five constructs (in addition to variables external to the model), which contain aspects that influence the actual use of the technology under consideration:

- External Variables (EV) represent contextual information from users and the environment.
- Perceived Usefulness (PU) is described as the extent to which users confirm that using the system improves their job performance.
- Perceived Ease of Use (PEU) is the extent to which users confirm that using the system would be free of corporeal and cerebral exertion.
- Attitude Towards Using the Technology (ATUT) relates to the users' perceptions of using the system, i.e., what is their attitude toward actually using the system (in all means).
- Behavioral Intention to use the Technology (BIT) denotes the users' intention of using the system.
- Actual Technology Use (ATU) assesses the system's performance and the extent to which it can meet the users' requirements [6].

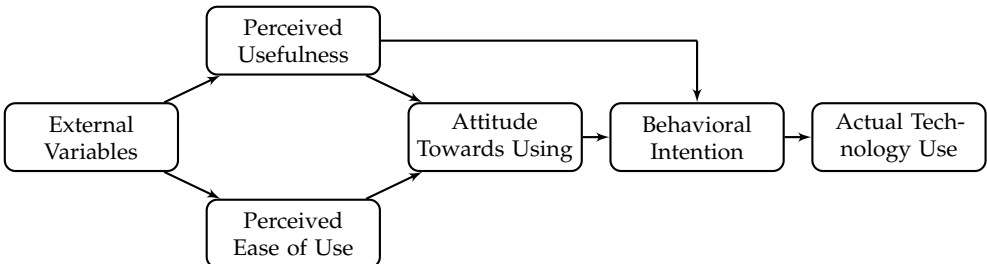

**Figure 1.** Technology Acceptance Model (adopted from [3]).

As shown in Figure 2, in this article, we extend the results presented in [2] in different ways. The first difference is related to the way data are considered. In [2], the answers to the questions could be selected from a 5-point Likert scale (ranging from strongly disagree (1) to strongly agree (5)). Therefore, [2] contemplated three different views of the data set by analyzing users' acceptance of Blackboard LMS specifically focusing on questions answered with scores "less than 3", "equal to 3", and "more than 3", whereas here, the data set is considered analyzed in its entirety.

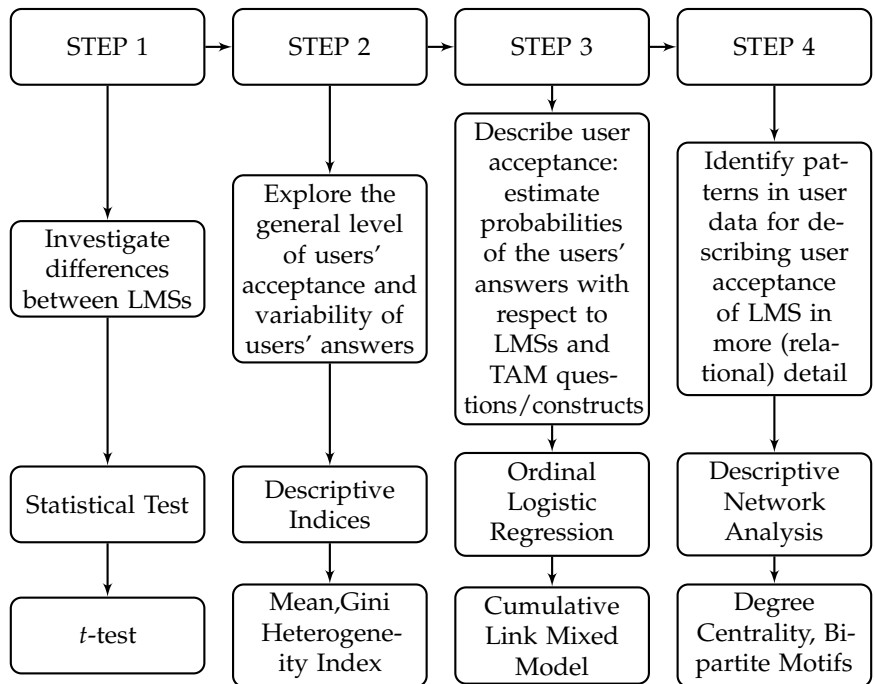

**Figure 2.** Summarizing the multi-level approach with the respective analysis methods.

We structure the analysis of the acceptance of the above LMSs according to the Technology Acceptance Model (TAM) [3], organizing this in two levels: questions and constructs. For both levels, first, we make use of descriptive statistics to explore the users' acceptance, showing general trends as well as variation between the answers provided by the users. Second, we employ a Cumulative Link Mixed Model, which describes user acceptance by estimating the probability of the users' answers considering the different LMSs and the questions/constructs. Third, we apply the descriptive network modeling to the obtained answers on TAM constructs (which is a similar technique as [2]), but for the two LMSs, i.e., Blackboard and Canvas) used on questionnaire data in which questions are organized in constructs. It represents the information provided by users for each construct of TAM as a network relating users based on their answers. Essentially, the network analysis approach identifies interesting patterns in the participant data. These patterns describe user acceptance of LMS in a more fine-grained manner (compared to the statistics approach). Overall, this enables different analysis levels using the respective targeted methods. Therefore, the main purpose is to demonstrate the data science techniques for data collection, processing, evaluation, and analysis, which are used in the context of the two LMSs.

Summing up, in this article, we target the following research questions, which extend those investigated in [2] (e.g., statistical and network analysis) to additional LMSs plus descriptive indices, e.g., the mean and the Gini heterogeneity index as well as the Cumulative Link Mixed Model, which enables a more comprehensive discussion:

1. What is the users' level of LMS acceptance for the following two LMSs: Blackboard versus Canvas?
2. What is the level of concordance for the users' acceptance?
3. How can we provide patterns for the users' LMS acceptance?

The rest of the article is structured as follows. In Section 2, we start with a brief introduction of the background related to LMSs and TAM. Next, Section 3 provides a description of the information coming from the questionnaire (for the two LMSs) and the methods that we employed for our investigations, e.g., the descriptive statistical indices and the statistical model. Additionally, in Section 4, we show the results of our analyses by comparing the data of the two LMSs (Blackboard versus Canvas). We provide results on both question and construct levels as follows:

1.  We examine the results of statistical tests to show if there are significant differences between the two LMSs.
2.  In addition, we make use of descriptive statistical indices, e.g., the mean and the Gini heterogeneity index, where the latter helps us to investigate the users' acceptance and the fluctuations amongst the answers provided by the users on a question-by-question basis.
3.  Furthermore, we use the Cumulative Link Mixed Model (which corresponds well to the ordinal data collected from the questionnaire on a 5-point Likert scale ranging from 1 to 5) to see the differences in the probability of answering questions/constructs while comparing the two LMSs.
4.  Finally, we apply descriptive network analysis approaches to provide patterns of users' LMS acceptance as well as the concordance in answering compared to [2].

This way, the Cumulative Link Mixed Model as well as network analysis can provide interesting insights for our data by measuring the effect of the questions and constructs plus their interactions with LMSs on the users' answers. This can be a complement to the general overview of the two LMSs, which is provided using descriptive statistical analyses. With respect to the TAM, the above approaches can be extended using descriptive network modeling, which results in patterns of users' LMS acceptance. Ultimately, we conclude with a summary and discussion of interesting future research directions.

## 2. Background

In this section, we first provide a brief overview of LMSs in general as well as some background information on the specific LMSs used in this study: Blackboard and Canvas. After that, we provide a short overview of the Technology Acceptance Model (TAM), which forms the basis for the design of the questionnaire, which is used to measure LMS acceptance.

### 2.1. LMS (Blackboard and Canvas)

An LMS is an electronic framework that allows for the creation, storage, reuse, management, and delivery of learning content. Most current LMSs are online, web-based systems that provide different interfaces for different functionalities or for different stakeholders. From the user perspective, an LMS provides learning content, such as lecture slides, instructional videos, and assessments, including exams or assignments (for online/offline use, and for local/distant learning). An LMS may also provide interaction with the instructor, e.g., by facilitating the submission of worked-out assignments, or through the use of forums, but also potentially with other users when working in groups [5]. From the instructor's perspective, an LMS allows for the easy distribution of learning material but also deals with user registration, user progress, and user results [7]. In general, it collects data to manage the learning and teaching process [1]. These data can be made available through reports that help instructors manage users better. As an example, they can organize users into groups to centralize reports and assignments. Using more advanced reports, it is also possible to follow the progress of large groups of users [8].

Various LMSs exist, with overall similar functionality, but also specific variations in user interfaces or in the level of functionality. In particular, Blackboard and Canvas are web-based LMSs that support both on-campus and online courses to plan, perform, and appraise learning processes. Blackboard is a popular LMS in the US, which is mostly aimed at colleges and universities (although other school types also use it) [9]. Canvas is another LMS specifically created for the academic environment and educational institutions [10].

The main differences between these LMSs can be found in two areas:

1.  Implementation and integration: Blackboard is originally designed for universities that want to host their own data. However, for universities that do not have enough resources or do not want to host the data, both Blackboard and Canvas allow for cloud deployment. Moreover, Canvas has a wide variety of tools to choose from, whereas

the Blackboard LMS only integrates with Dropbox, PowerSchool, and OneDrive (although these functionalities may change over time).

2.  Features: The two LMSs all have basic functionality: namely, Blackboard and Canvas both have a number of features in common, such as multi-user support, configurable learning portals, user-friendly design, and powerful user management capabilities. However, there are differences in additional features. For example, Blackboard users have to purchase modules that allow for specialized collaboration, such as the web conferencing function, whilst one primary feature of Canvas is the use of video as a source of content and collaboration.

### 2.2. Technology Acceptance Model (TAM)

As mentioned above, the TAM model describes that once users are provided with a new technology, several factors can influence their decision to use this technology. This process takes place within a certain environment, which is described using EV. These can be social factors (e.g., facilitating conditions, skills, and language), cultural factors (such as the perceived effect of using the technology within a social group), or political factors (e.g., the influence of technology on a political crisis). Within this environment, the directly important factors are PU and PEU. As improving the PU and PEU will also lead to an improvement of the ATUT and the BIT, developers need to realize the importance of the perceived system's usefulness and its ease of use [11]. The ATUT relates to the user's perception of the desirability of using the system, whereas BIT is the likelihood of the user actually using the system [12]. The ATU is now directly affected by the user's BIT.

Although numerous models have been proposed to describe the acceptance of systems, TAM describes this from a situational perspective and as such fits well to describe LMSs and e-learning [13–15]. Most other models aim to provide a detailed account, but they typically specialize due to the added complexity. For example, TAM2 [6] and the unified theory of acceptance and use of technology (UTAUT) [16], which are direct extensions of the standard TAM, focus on specifying new variables describing the EV in more detail. Alternatively, the valence model has a major focus on organizational aspects [17].

TAM has also been extended to include other types of information. For example, these models include two types of perceived usefulness (short-term and long-term) [18] or add a construct of compatibility [19]. The TAM3 version [20] includes constructs describing trust and risk. In addition, [21] examined individual acceptance and website usage and added two new structures to TAM: the value of perceived fun and the appeal of perceived presentation. Ref. [22] added playfulness constructs to analyze the World Wide Web acceptance. Several other publications show the usefulness of (extensions of) TAM in specific situations, such as the online shopping acceptance model (OSAM) to study online shopping behavior [23], whereas [24] used TAM to understand RFID acceptance and [25] investigated mobile service acceptance with perceived usefulness as the most important indicated factor.

Some studies relate the TAM to psychological models such as the Theory of Planned Behavior (TPB) and a decomposed TPB model. For instance, the study of [26] applies these models in Hong Kong's healthcare setting. The results highlight the superiority of TAM over TPB in explaining physicians' intention to use telemedicine technology.

More relevant to our study, some previous research investigated the impact of demographics (e.g., gender, age, field of study) on TAM constructs. The results show that there is not a substantial connection between perceived usefulness and users' demographics [27,28]. However, older users seem to better comprehend the usefulness of the system under consideration [29]. Additionally, the users' level of education seems to play a crucial role in the perceived usefulness [30,31]. Similar results have been obtained for other constructs, e.g., BIT, ATUT, and ATU [29,32]. Previous work emphasized statistical descriptions within the model [11,33,34]. Similarly, [35] confirmed the relationship amid PEU, PU, ATUT, and the overall impact on BIT. In addition, they showed that the external variables, e.g., job relevance, have a robust association with TAM constructs such that job relevance can have

a positive effect on LMS usefulness (PU). Furthermore, [36] measured the usability of three open-source LMSs: Moodle, ILIAS, and Atutor. According to the results, Moodle, due to the attractive interface, was most easy to use compared to the other two LMSs. Considering this overview, we believe that TAM is a good fit and a proper framework to predict the behavioral intention to use a system.

Summarizing, TAM delivers a concrete (and simple) model to define users' acceptance of novel technologies and can be successfully applied in an LMS context. The model consists of five constructs and their interconnections. As such, they describe how we can expect users to accept LMSs, e.g., Blackboard and Canvas. To investigate acceptance, TAM provides us with specific areas that influence acceptance, which can be used to ask the users specifically about their perception of these areas. This means that the model can help us to structure a questionnaire for the investigation into the acceptance of the LMSs. This provides information on two levels: the individual questions as well as their organization in the TAM constructs. Exactly how this is accomplished is described in Section 3 and 4.

### 2.3. Statistical Analysis

To investigate the LMS users' acceptance as measured by the questionnaire structured according to TAM, we perform some statistical analyses on the data. This allows us to observe the acceptance per system and to compare the considered LMSs. For this, we use a statistical model based on the questionnaire values (per question and construct). In addition, we look at the Gini heterogeneity index [37], which has been applied to questionnaire data in order to evaluate whether answers are concentrated mostly in only one category (i.e., potential answer) or whether they are mostly equally distributed along all answers of a question. In other words, this evaluates the level of accordance between individuals.

Furthermore, a mixed model can tell us whether there is a difference in the probability of selecting a value (answer) for each variable (question) when comparing the datasets or not. This means that by allowing for an interaction between question and interface, this model makes a big assumption; i.e., the probability of each answer can be modeled as a linear combination of the likelihood of each answer under each interface as well as the likelihood of each answer for each question [38].

### 2.4. Network Analysis (Centrality and Motifs)

Network analysis methods in general are almost exclusively used to analyze relational data. In relational data, the links (relationships) between actors (users, objects, companies, etc.) matter to explain some phenomena in the data. This type of relational data can be modeled using a network, which consists of a set of nodes (also called vertices), which represent the actors or objects, and a set of edges (or arcs or links), which describe the relationships. Networks, which can also be represented as graphs, can be *directed* if the edges only run in one direction (from one node to another) or *undirected* or *biunivocal* if the edges run in both directions between two nodes. Additionally, if the edges of a network merely depict the absence/presence of a relationship between the nodes, it is called *unweighted*, whereas when the strength of a link is provided, the network is called *weighted*. A specific type of network is the *2-mode* or *bipartite* network (in contrast to the usual one-mode network). These networks are made up of two distinct types of nodes, and the edges only exist between the different types (connecting one of each type). Networks may also be even more complicated. For example, multi-layer networks comprise multiple, dissimilar kinds of nodes and edges. This allows for many global systems (e.g., social networks) to be represented as networks [39].

The advantage of viewing data as a network is that a range of networks analysis methods may be used to extract additional information (compared to "regular" statistical analysis methods, i.e., descriptive statistics and inferential statistics) as these methods can focus on the inherent relational aspects of the data, which may provide additional information that is closely related to the research questions. Ref. [40] discussed the use of network approaches on questionnaire data; an additional discussion on the different

network analyses can be found in [41], stressing that the network analysis approach focuses on properties of pairs of users (i.e., dyadic relationships), thus also providing information on a more relational level.

Furthermore, [42] proposed a network analysis model from a Likert-scale survey. They created a bipartite network from users' answers based on Likert-scale selections. To present the number of users selecting similar answers, they used the edge weights. In other words, using the edge weight in the network, the similarity of the Likert-scale selections could be presented. They were also able to find the advantages of this approach by comparing network analysis and principal component analysis (PCA) so they construct a meaningful network based on the similarities and differences of answering between users. According to [42], this proposed methodology can be generalized to any set of Likert-scale surveys for network-based modeling. Likewise, in [2], the users' answers were based on Likert-scale selection (from 1 to 5). We thus considered three different views of the data set, focusing on questions answered with scores "less than 3", "equal to 3", and "more than 3". We applied the frequency distributions through descriptive network analysis tools, namely degree centrality and bipartite motifs.

Centrality measurement provides information about the importance of a node in the graph. There are four main centrality criteria: degree centrality, closeness centrality, betweenness centrality, and eigenvector centrality. Degree centrality for a node is basically defined via its degree; thus, the greater the degree, the more central the node will be. In other words, a high degree of centrality implies that a node includes more connections than the average graph. There are two types of criteria for directed graphs: in-degree and out-degree. The former is considered to be the number of edges that point toward the node and the latter is considered to be the number of nodes directed away from the given node [40]. The closeness measures the average distance between one node and other nodes, so the more central a node is, the closer the node is to the other nodes. The betweenness is defined as the number of shortest paths in which a node is located which is commonly used to see the information flow in the graph. The higher the betweenness, the more information flows within the graph. The eigenvector is about a node's relative impact within the network or how connected a node is to other highly connected nodes [39].

Whereas the basic network analysis metrics, e.g., degree centrality provides information of the overall structure of the network, it is possible to gain further insight into the structure of the network using more advanced, structural analyses by applying a motif extraction approach, c.f., [43,44].

Motifs are particular subgraphs of bipartite networks considered as the basic "building blocks" of networks that include both types of nodes [45]. As shown in Figure 3, you may observe two nodes in the top set (A) and three nodes in the lower set (B) in motifs 14, 15, and 16. The product of binomial coefficients, selecting two nodes from A and three nodes from B, thereby gives the maximum number of node combinations that could exist in these patterns: $\binom{A}{2}\binom{B}{3}$ [46]. With respect to our data set, this indicates that motif configurations include one or two users and many questions (3, 4, or 5) or many users (3, 4, or 5) and one or two questions. They represent patterns of questions receiving the same answer from a user or patterns of users responding to a question in the same way.

In general, the sizes of motifs can be varied from two to six nodes (larger is possible, but depending on the size of the bipartite network, these may hardly ever occur) and include all the isomorphism classes. Bipartite motifs can be used in different ways, for example, to compute the number of repetitions of different motifs in a network [47]. Likewise, they can be helpful while quantifying the role of nodes in a group by counting the number of nodes that appear in various positions of motifs [48]. The benefit of motifs is that with respect to traditional indices, they are much more sensitive to changes within the network. This means that while many network configurations have similar index values, a small number of network configurations have the same motif structure [43]. Furthermore, bipartite motifs are well suited to represent the relationships (answers) between one user and a group of questions, a group of users and one question, or a group of users and a group of questions.

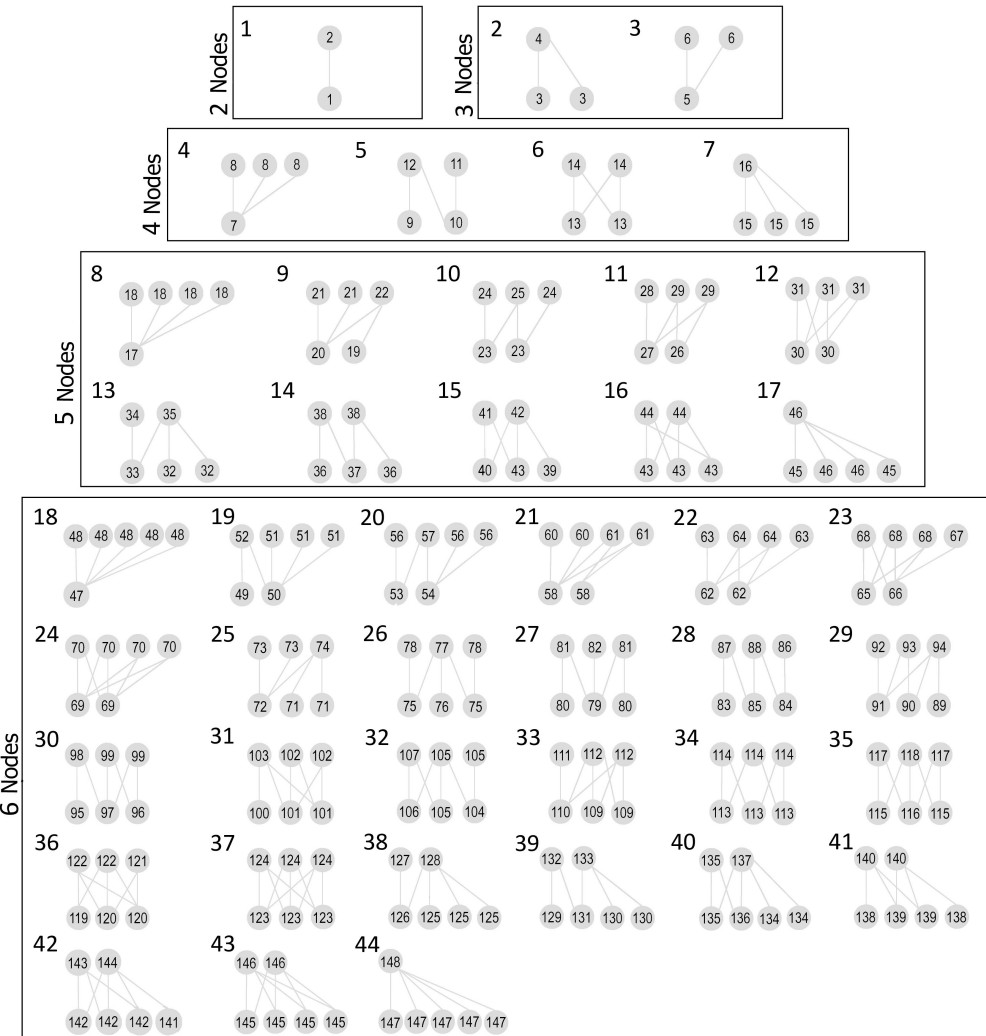

**Figure 3.** All possible bipartite motifs from two to six nodes.

## 3. Methodology

To collect information on the acceptance of LMSs by users, we used a questionnaire [49]. The types of questions and their answers allowed us to perform quantitative analyses.

As explained in Section 1, at the question and the construct level, we investigate statistical tests. This helps to see whether there are significant differences between the LMSs. Furthermore, we make use of descriptive statistical analysis (e.g., mean and Gini heterogeneity index) as well as statistical modeling (Cumulative Link Mixed Model), where the former shows the level of user acceptance and their concordance, while the latter helps us to estimate the answers given by the users considering the distinction between LMSs, questions/constructs, and their interaction. Finally, we apply descriptive network modeling (e.g., degree centrality and bipartite motifs) at the construct level to examine the variability of users' answers and the patterns of user satisfaction in different networks accordingly.

Below, we provide an overview of the approach. First, we describe the design of the questionnaire and, second, we provide an overview of the methods used for the descriptive as well as the statistical and network-based modeling analyses.

### 3.1. Material

For the current study, we collected data on the acceptance of two LMSs at one university using the same questionnaire. We will first provide information on the questionnaire

that was used followed by a short description of the data set for each LMS. Based on the answers to the questions, we provide a short analysis of the reliability of the questionnaire.

The questionnaire consists of two parts. The first part is a small set of demographic questions. The second part consists of 30 questions taken from [49], which together measure the five TAM constructs. Perceived usefulness (PU) is measured in questions 1–6, perceived ease of use (PEU) is measured in questions 7–11, behavioral intention (BIT) is measured in questions 12–15, attitude toward using (ATUT) is measured in questions 16–23, and actual technology use (ATU) is measured in questions 24–30. Table 1 provides an overview of the questions. Note that the answers to the questions in the second part can be selected from a 5-point Likert scale (ranging from strongly disagree (1) to strongly agree (5)).

**Table 1.** List of questions in the questionnaire (where "LMS" is replaced by the name of the LMS under consideration).

| Q | Description |
|---|---|
| **PU** | |
| 1 | LMS helps me to increase my learning productivity |
| 2 | LMS helps me to find the course materials |
| 3 | LMS helps me to submit the assignments |
| 4 | LMS increases my academic performance |
| 5 | LMS helps me in the learning process |
| 6 | LMS helps me to ask and discuss some topics with the lecturer |
| **PEU** | |
| 7 | LMS is easy to operate |
| 8 | LMS uses understandable language |
| 9 | LMS uses the appropriate background color and font |
| 10 | LMS has a systematic menu |
| 11 | LMS is accessible from within and outside of the university |
| **BIT** | |
| 12 | I have the intention to use LMS every day |
| 13 | I have the intention to check the latest materials on LMS |
| 14 | I have the intention to check my grade through LMS |
| 15 | I have the intention to encourage my fellow users to use LMS |
| **ATUT** | |
| 16 | I use LMS without any compulsion from anyone |
| 17 | I need LMS |
| 18 | I am happy when I use LMS |
| 19 | Using LMS to submit the assignment is an innovative idea |
| 20 | Using LMS to download the course materials is an innovative idea |
| 21 | Using LMS to discuss with lecturer/fellow users is a positive idea |
| 22 | Using LMS is a good and wise decision |
| 23 | I am going to encourage my fellow users to use LMS |
| **ATU** | |
| 24 | I use LMS to support the learning activities |
| 25 | I always access LMS every day |
| 26 | I get the course materials from LMS |
| 27 | I download and upload assignments through LMS |
| 28 | I use LMS to check my grades |
| 29 | I am satisfied using LMS |
| 30 | I tell my fellow users about my satisfaction using LMS |

The data of the first LMS, Blackboard, describes answers to the questions in the questionnaire from 51 pre-master LMS users (out of a total of 118 people registered as pre-master students in the full academic year; note that the pre-master program is only one semester, but student registration is captured per academic year) from the School of

Humanities and Digital Sciences School at Tilburg University (Tilburg, The Netherlands). These were collected during the spring (i.e., last) semester in the academic year 2018–2019. Of the 51 users, 25 (49.0%) were female and 26 (51.0%) were male. The age of the users was distributed as follows: 44 (86.3%) were in the age range between 20 and 30, six (11.8%) were in the age range between 31 and 40, and one (2.0%) user was over 40 years old. This part of the data set has also been used in a previous study [2].

For the second LMS, Canvas, answers from 49 pre-master users (out of a total of 95 people registered as pre-master students that academic year) from the School of Humanities and Digital Sciences at Tilburg University were collected during the fall (i.e., first) semester in the academic year 2019–2020. Out of 49 users, 27 (55.1%) were female and 22 (44.9%) were male. Most users (46, 93.9%) were in the age range between 20 and 30, two (4.1%) were in the age range between 31 and 40, and one (2.0%) was over 40 years old.

Note that both Blackboard and Canvas users have relatively similar experiences of LMS (e.g., submit the assignments, check the course materials/grades, and discuss with lecturer/fellow users through the LMS).

To illustrate the reliability of the questionnaire (both for the overall questionnaire as well as for the individual TAM constructs separately), we compute the Average Variance Extracted (AVE), Composite Reliability (CR), R-squared ($R^2$), and the respective Cronbach $\alpha$s [50]. AVE is used to measure the variance degree for a construct; the values of greater than 0.5 indicate that the reliability of the result is more acceptable. CR is an indicator to measure the internal integrity in which the values should be higher than 0.6. $R^2$ measures the proportion of the variance for each construct such that the values greater than zero are acceptable. Table 2 shows the different values for the two LMSs. Previous research has already shown the reliability of the questionnaire with Cronbach $\alpha$-values above 0.7 [49], and here, we observe similar results with Cronbach $\alpha > 0.7$ for all constructs in LMSs. The overall Cronbach $\alpha$s for the LMSs are larger than 0.9, which means that the questionnaire's reliability is excellent. Additionally, the reliability of the questionnaire for each construct is considered acceptable.

**Table 2.** Average Variance Extracted (AVE), Composite Reliability (CR), $R^2$, and Cronbach $\alpha$ of the results from the questionnaire for each TAM construct and total (combined) LMS (for Blackboard and Canvas).

| Blackboard | AVE | CR | $R^2$ | $\alpha$ |
|---|---|---|---|---|
| PU | 0.401 | 0.763 | 0.145 | 0.822 |
| PEU | 0.477 | 0.972 | 0.106 | 0.706 |
| BIT | 0.552 | 0.831 | 0.142 | 0.701 |
| ATUT | 0.859 | 0.687 | 0.265 | 0.855 |
| ATU | 0.352 | 0.754 | 0.149 | 0.836 |
| Total | 0.128 | 0.737 | 0.559 | 0.942 |
| Canvas | AVE | CR | $R^2$ | $\alpha$ |
| PU | 0.414 | 0.776 | 0.167 | 0.813 |
| PEU | 0.721 | 0.927 | 0.156 | 0.891 |
| BIT | 0.494 | 0.791 | 0.236 | 0.744 |
| ATUT | 0.541 | 0.903 | 0.465 | 0.876 |
| ATU | 0.578 | 0.904 | 0.238 | 0.866 |
| Total | 0.161 | 0.789 | 0.417 | 0.944 |

*3.2. Statistical Analyses*

To investigate the LMS users' acceptance as measured by the questionnaire structured according to TAM, we perform statistical analysis on the data, using descriptives and modeling methods. This allows us to observe the acceptance per system and to compare the two LMSs considered in this study at the question and construct level.

For the descriptive statistical tools, we use the mean, standard deviation, and the Gini heterogeneity index [37] for both the question and construct level. The Gini heterogeneity

index indicates how far answers in Likert-scale questionnaire data are concentrated mostly in only one specific answer (i.e., value) or whether they are more equally distributed over all answers to a question. In other words, this evaluates the variability of each question, namely the level of accordance among individuals. While for the question level, the Gini heterogeneity index is a natural index, due to the reliability observed in Cronbach's $\alpha$, we can also apply it at the construct level. Each construct is then considered to be a unique block (same distributions of questions) and can be vectorized.

The Gini heterogeneity index was proposed by Corrado Gini [37,51] as one instantiation of statistical inequality measures. Here, we investigate a specific ordinal variable with its associated set of categories with the Gini heterogeneity index $G$ defined as:

$$G = \frac{m}{m-1}\left(1 - \sum_{j=1}^{m} p_j^2\right),$$

(1)

where $m$ is the number of categories described by the ordinal variable (in our case $m = 5$ as we deal with 5-point Likert scales), and $p_j$ is the relative frequency of each category, $j = 1, \ldots, m$.

If we observe only occurrences in one category (i.e., the relative frequency for one category $p_j = 1$), then the heterogeneity is minimal ($G = 0$) (and for the questionnaire data, the concordance of users' answers is maximal). If the relative frequencies are the same for all categories $p_j = \frac{1}{m}$ for $j = 1, \ldots, m$, then the heterogeneity is maximal ($G = 1$), but in the questionnaire case, the concordance in answers between users is at the minimum.

The Cumulative Link Mixed Model (CLMM) [38,52,53] is a statistical modeling approach that can tell us whether there is a question or construct effect on the probability of selecting a value (answer) by a user when comparing the LMSs. Similar to the mean and Gini heterogeneity index, we apply this model to both the question and construct levels.

We can organize the questionnaire answer data for different users in a table with five columns. Each row represents the answer to a particular question by a particular user. Additional information on the LMS and the corresponding construct is also added. Therefore, we can represent the LMS data in a matrix with users as rows and questions (per construct) as columns; however, to apply the CLMM, it is required to re-organize the data.

The main goal of the new organization is to put the two LMSs together, considering that in each row, one answer given by a user to a question from one LMS is repeated (i.e., one user will be repeated 30 times in 30 rows). The final matrix is composed of four variables: the first is the answer given by the user, the second indicates the LMS, the third is the corresponding question, and the fourth is the corresponding construct. We can thus model the probability of answers by including LMSs and the question/construct effect as well as the interaction between them (i.e., LMS and question/construct) by adding a grouping factor effect for the users. Allowing for interaction between the question and the LMS, this model makes an important assumption (i.e., the probability of each answer (1–5) is modeled as a linear combination of the likelihood of each answer under each LMS as well as the likelihood of each answer for each question/construct).

### 3.3. Network Analysis

Before we can apply network analysis measures to investigate relational information in the data, the data set will need to be converted into a network. Networks can be represented as graphs, which are typically used to visualize them, but they can also be represented using adjacency matrices (which are called affiliation matrices for two-mode networks) or using edge lists. Adjacency matrices consist of $n$ rows and $n$ columns that are labeled with identifiers for the $n$ nodes, and each entry $(i, j)$ in the $n \times n$ matrix (each cell) represents the value of the link between the two nodes $i$ and $j$. Unweighted graphs may have true/false or 1/0 values in their matrices, whereas weighted graphs may contain real numbers indicating the weights of edges in the respective entries of the adjacency matrix. In the case of questionnaire data, there are three types of information: users, questions, and

answers given by users to the questions. From this, we can construct a weighted bipartite network for each TAM construct with the set of users *U* and the set of questions *Q* as nodes. The edges between elements from the sets *U* and *Q* indicate that a user (from *U*) provides an answer to a question (from *Q*) and the answer information can then be encoded as a weight on each edge. The same as previous study [2], for each TAM construct, we create three unweighted bipartite graphs with the same nodes as the original graph (representing users and questions), but only those edges that adhere to a particular criterion: edges that have the answer (weights). 1. less than three, 2. equal to three, 3. more than three (see Figure 4).

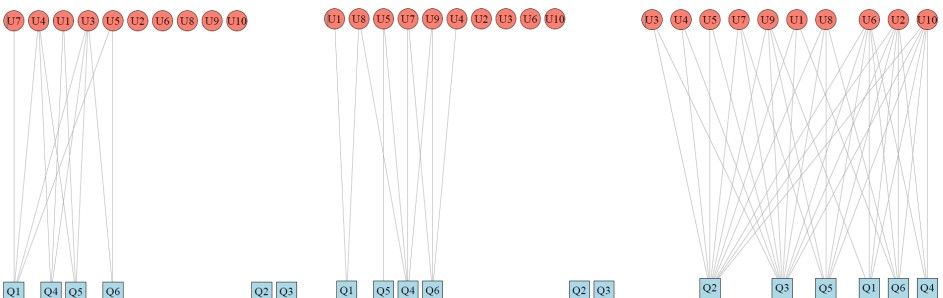

**Figure 4.** From the left to the right, an example of "less than 3", "equal to 3", and "more than 3" networks (User/Question) within the PU construct of the Blackboard questionnaire.

### 3.3.1. Degree Centrality

For network characterization and the identification of interesting properties, we can apply descriptive network analysis methods or apply more complex models. These analyses provide information on the overall shape or other properties of the network. Here, we are interested in studying and comparing the users' LMSs acceptance at the construct level. This is well measured by the variability of the users' degree centrality distribution in their unipartite weighted networks, which are obtained as projections from the constructed bipartite networks.

Considering an adjacency matrix *A* describing the LMS acceptance results for one construct, with the main diagonal equal to zero (so that there are no self-loops for the set of nodes), the formulation of the weighted degree is defined as

$$d_i = \sum_{j=1}^{n} a_{ij},$$

(2)

where *n* is the number of (participant) nodes and $a_{ij}$ represents the entries of the adjacency matrix *A*. The degree centrality $d_i$ can be seen as the level of concordance of each participant with respect to the other participants. The average value of the degree distribution is the average level of concordance between participants, so it describes a similar measure to the Gini heterogeneity index but is now on the construct level. In addition, the variance of the composite degree centrality indicates the consistency of users' acceptance within the network. Focusing on it and allowing for comparison between constructs and LMSs, we make use of the normalized coefficient of variation.

### 3.3.2. Bipartite Motifs

As mentioned above, motifs are specific subgraphs of bipartite networks that include two sets of nodes. Given this, we represent our data set as a bipartite network with users as a set of nodes *U* and questions as a set of nodes *Q* to indicate the patterns between users and questions.

For instance (as displayed in Figure 3), you may observe two nodes in the top set (A) and three nodes in the lower set (B) in motifs 14, 15, and 16. The product of binomial coefficients, selecting two nodes from A and three nodes from B, thereby gives the maximum

number of node combinations that could exist in these patterns: $\binom{A}{2}\binom{B}{3}$ [46]. With respect to our data set, this indicates that motif configurations include one or two users and many questions (3, 4, or 5) or many users (3, 4, or 5) and one or two questions. They represent patterns of questions receiving the same answer from a user or patterns of users responding to a question in the same way. Given a bipartite network, it is now possible to identify and count the different motifs. Here, we follow the approach first presented in [2], where we record the following information per motif: (1) motif ID, (2) the number of nodes in the motif, (3) the absolute frequency of each motif, (4) the relative frequency of each motif as a proportion of (a) the total number of motifs with a specific configuration in the network, and (b) the possible number of motifs with the same configuration.

The complete set of motifs—according to the discussion above—is shown in Figure 3; please see [43] for a detailed discussion. As motifs describe the actual structure of a bipartite network, they provide more specific information compared to the "standard" network metrics. In fact, networks that may show similar values for the basic network analysis metrics, in reality, show different configurations [43]. Counting the occurrences of the motifs and calculating their relative frequencies can show the differences in the structures of the unweighted bipartite networks defined above along each TAM construct of the investigated LMSs.

## 4. Results

As mentioned in previous sections, the analysis of the LMS acceptance (represented as the answers to the questions in the questionnaire) can be performed on two levels: per question and per construct. We will first consider the different types of analysis (both descriptives and the CLMM) on the question level. Next, the same analyses plus descriptive network analysis are performed on the construct level.

### 4.1. Descriptive Analysis (Question Level)

Table 3 contains the mean, standard deviation, and Gini heterogeneity index values for each of the questions for each of the LMSs. The mean values of the different questions related to the level of users' LMS acceptance. The standard deviation provides a measure of the spread of the values provided by the users. The Gini heterogeneity index shows the variability of answers taking into account the users' concordance and their agreement on the answer to each question. It is important to note that a standard deviation may be relatively large if some people provide extreme answers (with respect to the mean), but this may still lead to relatively high concordance (according to the Gini heterogeneity index) if multiple users do this.

Investigating the results in Table 3, we see that on average for all questions combined, Canvas shows higher scores (3.9), although the results for Blackboard are not far behind (3.8). What may be more interesting is the variation in the scores. For this, we can take a look at the standard deviations of the scores. Canvas shows a smaller standard deviation (0.8) than Blackboard (0.9), indicating that there is a relatively smaller spread in the results for Canvas.

Another way of looking at the variation is according to the Gini index. The Gini index of zero corresponds to the maximum of concordance, and a Gini index of one describes a perfect distribution over all possible answers. The Gini index of Canvas users is slightly lower (0.110) than that of Blackboard (0.162), indicating that Canvas users are more consistent in providing their answers. To investigate whether there are significant differences between the assignment of scores to each of the questions, we applied a *t*-test to each of the questions. Note that by applying the *t*-test per question, we essentially assume independence between the questions (which we know is not true). We do not apply any correction for this, since the test is really applied in order to obtain a sense of where possible differences may be. The CLMM (described in Section 4.2) will provide a more fine-grained insight. The results of the *t*-tests can be found in Table 3. This shows that for all questions, we do not identify any significant differences between the two systems ($p > 0.5$).

**Table 3.** Mean (*M*), standard deviation (*SD*), and Gini heterogeneity index (Gini) values for each TAM question for Blackboard and Canvas LMSs. In addition, the *t*-values of the *t*-tests comparing the results per question between the systems and the corresponding *p*-values are provided. At the bottom of the table, the mean and standard deviations over all questions are provided.

| Q | Blackboard | | | Canvas | | | t | p |
|---|---|---|---|---|---|---|---|---|
| | *M* | *SD* | Gini | *M* | *SD* | Gini | | |
| 1 | 3.569 | (1.025) | 0.148 | 4.061 | (0.556) | 0.063 | 0.053 | 0.959 |
| 2 | 4.549 | (0.610) | 0.062 | 4.469 | (0.581) | 0.065 | 0.048 | 0.963 |
| 3 | 4.471 | (0.612) | 0.065 | 4.388 | (0.786) | 0.087 | 0.050 | 0.961 |
| 4 | 3.255 | (0.935) | 0.153 | 3.571 | (0.707) | 0.101 | 0 .065 | 0.949 |
| 5 | 3.509 | (1.007) | 0.152 | 3.837 | (0.850) | 0.119 | 0.077 | 0.940 |
| 6 | 3.353 | (1.146) | 0.179 | 3.694 | (0.918) | 0.134 | 0.078 | 0.939 |
| 7 | 3.569 | (1.153) | 0.173 | 4.306 | (0.742) | 0.087 | 0.066 | 0.949 |
| 8 | 4.039 | (0.774) | 0.087 | 4.388 | (0.606) | 0.069 | 0.050 | 0.961 |
| 9 | 3.745 | (0.891) | 0.119 | 4.408 | (0.674) | 0.077 | 0.057 | 0.956 |
| 10 | 3.353 | (1.092) | 0.178 | 4.142 | (0.979) | 0.123 | 0.081 | 0.937 |
| 11 | 3.882 | (1.107) | 0.145 | 4.327 | (0.718) | 0.083 | 0.063 | 0.951 |
| 12 | 3.961 | (0.871) | 0.110 | 4.000 | (0.913) | 0.122 | 0.066 | 0.949 |
| 13 | 4.196 | (0.749) | 0.088 | 3.878 | (0.881) | 0.119 | 0.061 | 0.953 |
| 14 | 4.471 | (0.504) | 0.056 | 3.898 | (0.918) | 0.126 | 0.056 | 0.957 |
| 15 | 3.412 | (1.043) | 0.167 | 3.571 | (0.913) | 0.135 | 0.085 | 0.935 |
| 16 | 3.922 | (0.997) | 0.126 | 3.816 | (0.858) | 0.118 | 0.064 | 0.951 |
| 17 | 4.118 | (0.791) | 0.095 | 3.816 | (0.858) | 0.118 | 0.060 | 0.953 |
| 18 | 3.039 | (0.871) | 0.149 | 3.673 | (0.747) | 0.106 | 0.064 | 0.950 |
| 19 | 3.235 | (1.106) | 0.187 | 3.429 | (0.957) | 0.149 | 0.095 | 0.927 |
| 20 | 3.431 | (1.171) | 0.187 | 3.408 | (1.019) | 0.161 | 0.010 | 0.919 |
| 21 | 3.941 | (0.785) | 0.094 | 3.837 | (0.825) | 0.109 | 0.056 | 0.957 |
| 22 | 3.922 | (0.771) | 0.104 | 4.143 | (0.764) | 0.098 | 0.065 | 0.950 |
| 23 | 3.412 | (0.984) | 0.158 | 3.673 | (0.899) | 0.128 | 0.079 | 0.939 |
| 24 | 3.941 | (0.810) | 0.105 | 3.878 | (0.807) | 0.106 | 0.061 | 0.953 |
| 25 | 3.725 | (0.939) | 0.121 | 3.653 | (0.903) | 0.129 | 0.059 | 0.954 |
| 26 | 4.451 | (0.503) | 0.057 | 4.265 | (0.729) | 0.089 | 0.051 | 0.961 |
| 27 | 4.471 | (0.504) | 0.056 | 4.204 | (0.841) | 0.103 | 0.053 | 0.959 |
| 28 | 4.490 | (0.543) | 0.059 | 3.918 | (0.862) | 0.117 | 0.056 | 0.957 |
| 29 | 3.784 | (0.966) | 0.138 | 4.184 | (0.783) | 0.099 | 0.073 | 0.943 |
| 30 | 2.863 | (1.077) | 0.204 | 3.571 | (1.041) | 0.155 | 0.088 | 0.932 |
| *M* | 3.803 | 0.878 | 0.162 | 3.947 | 0.821 | 0.110 | | |
| *SD* | 0.468 | 0.209 | 0.134 | 0.312 | 0.122 | 0.025 | | |

To investigate the relationship between the mean answers of the questions and their corresponding Gini heterogeneity indices (which describe the internal consistency of the answers for each question), we plot the results of Table 3 in Figure 5. Here, the *x*-axis portrays the mean answer and the *y*-axis depicts the Gini heterogeneity index. Moving to the right on the *x*-axis, we find more positive mean answers for the questions. Going down on the *y*-axis shows lower heterogeneity index values, which corresponds to users giving more similar answers. Note that the color of the points in the figure represents the LMS system, and the shape of the points denotes the TAM construct for each question, which will be discussed later. The figure illustrates several interesting properties. First, we see that the mean scores for the different questions from Blackboard are slightly lower (more to the left in the figure) compared to the Canvas system. These scores for both LMSs, however, are all relatively close together; i.e., the spread of the scores tends to be very similar for most questions.

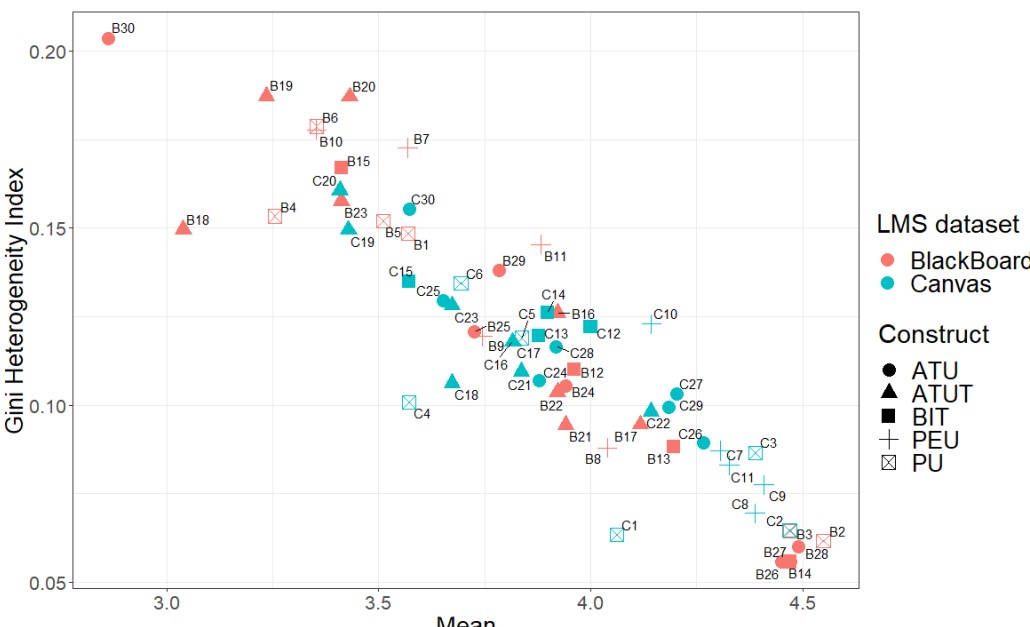

**Figure 5.** Descriptive summary from the answers to the TAM questions and the variability between the answers per question for Blackboard and Canvas.

Second, for both systems, there is a trend to have higher Gini values if the scores are lower. This can (partially) be expected. If we consider the extremes, e.g., mean scores close to either the maximum or minimum values, then this means that users generally need to select the extreme value (otherwise, the mean would be further away from the extreme value). Similarly, the standard deviation would be small. If we now consider mean scores closer to the middle of the possible scores, there may be more variation in the scores to end up with such a mean. Still, one may expect that if some Blackboard users, with mean scores around two, give a score of two to the questions, then the Gini index would be lower, indicating higher internal coherence. In this case, the mean scores for the Blackboard users are established due to some variation in the scores provided by the users, which is also reflected in the higher standard deviations in Table 3.

The figure also indicates a few outlier questions. Some Blackboard questions show lower scores. In particular, question 30 has a low score (and high Gini value). This is the last question in the questionnaire (see Table 1) and deals with whether a user would tell other users about their satisfaction. The corresponding values for Canvas are also on the lower side, but these are not as extreme. The other questions that could be seen as outliers (low values compared to Canvas) are mostly related to ATUT questions (18, 19, 20) and PU (4, 6). These will be discussed below (in Section 4.3).

### 4.2. Cumulative Link Mixed Model (Question Level)

To investigate the relationships between LMSs and the answers to the questions, we build a Cumulative Link Mixed Model. This model predicts the most likely answer (out of values 1–5) for each question while considering the LMS. The model relies on a linear combination of the weights of the LMS and the question. This model makes the assumption that the likelihood of an answer can be modeled as a linear combination of information from the LMS and the question (allowing for an interaction between the question and LMS).

The model fits the answers given by the participants based on the LMS and question variables where we also look at potential interactions. The participant is incorporated in the model as a random variable. The weights of the LMS and question variables can be found in Table 4 with Blackboard and Q1 as a reference. The interaction effects between LMS and questions can be found in Table 5 (due to space limitations).

**Table 4.** Weights and *p*-values for the two variables (LMS and questions) indicating the significant influences of the values for the variables in the Cumulative Link Mixed Model, where "***" indicates $p < 0.001$, "**" indicates $p < 0.01$, "*" indicates $p < 0.05$, and "." indicates $p < 0.1$.

| Coefficients | Weight | p | | Coefficients | Weight | p | |
|---|---|---|---|---|---|---|---|
| Canvas | −0.611 | 0.022 | * | Q16 | 0.322 | 0.263 | |
| Q2 | 2.363 | <0.001 | *** | Q17 | 0.473 | 0.100 | |
| Q3 | 2.105 | <0.001 | *** | Q18 | −1.274 | <0.001 | *** |
| Q4 | −1.187 | <0.001 | *** | Q19 | −1.303 | <0.001 | *** |
| Q5 | −0.473 | 0.093 | . | Q20 | −1.024 | <0.001 | *** |
| Q6 | −0.750 | 0.009 | ** | Q21 | 0.198 | 0.489 | |
| Q7 | 0.538 | 0.065 | . | Q22 | 0.598 | 0.034 | * |
| Q8 | 1.282 | <0.001 | *** | Q23 | −0.810 | 0.004 | ** |
| Q9 | 0.875 | 0.002 | ** | Q24 | 0.286 | 0.312 | |
| Q10 | −0.018 | 0.950 | | Q25 | −0.356 | 0.203 | |
| Q11 | 1.086 | <0.001 | *** | Q26 | 1.745 | <0.001 | *** |
| Q12 | 0.486 | 0.092 | . | Q27 | 1.735 | <0.001 | *** |
| Q13 | 0.752 | 0.009 | ** | Q28 | 1.284 | <0.001 | *** |
| Q14 | 1.242 | <0.001 | *** | Q29 | 0.541 | 0.057 | . |
| Q15 | −0.926 | <0.001 | *** | Q30 | −1.554 | <0.001 | *** |

**Table 5.** Weights and *p*-values between two variables (LMS and questions) indicating the significant influences of the values for the variables in the Cumulative Link Mixed Model, where "***" indicates $p < 0.001$, "**" indicates $p < 0.01$, "*" indicates $p < 0.05$, and "." indicates $p < 0.1$.

| Coefficients | Weight | p | | Coefficients | Weight | p | |
|---|---|---|---|---|---|---|---|
| Canvas-Q2 | 0.803 | 0.007 | ** | Canvas-Q17 | 1.060 | <0.001 | *** |
| Canvas-Q3 | 0.705 | 0.018 | * | Canvas-Q18 | −0.188 | 0.487 | |
| Canvas-Q4 | 0.260 | 0.341 | | Canvas-Q19 | 0.359 | 0.198 | |
| Canvas-Q5 | 0.214 | 0.447 | | Canvas-Q20 | 0.655 | 0.021 | * |
| Canvas-Q6 | 0.291 | 0.311 | | Canvas-Q21 | 0.716 | 0.012 | * |
| Canvas-Q7 | −0.459 | 0.115 | | Canvas-Q22 | 0.212 | 0.453 | |
| Canvas-Q8 | 0.053 | 0.855 | | Canvas-Q23 | 0.236 | 0.395 | |
| Canvas-Q9 | −0.479 | 0.097 | | Canvas-Q24 | 0.716 | 0.011 | * |
| Canvas-Q10 | −0.616 | 0.033 | * | Canvas-Q25 | 0.744 | 0.007 | ** |
| Canvas-Q11 | −0.023 | 0.938 | | Canvas-Q26 | 0.934 | <0.001 | *** |
| Canvas-Q12 | 0.484 | 0.092 | . | Canvas-Q27 | 1.023 | <0.001 | *** |
| Canvas-Q13 | 1.001 | <0.001 | *** | Canvas-Q28 | 1.568 | <0.001 | *** |
| Canvas-Q14 | 1.526 | <0.001 | *** | Canvas-Q29 | −0.005 | 0.986 | |
| Canvas-Q15 | 0.407 | 0.144 | | Canvas-Q30 | −0.342 | 0.220 | |
| Canvas-Q16 | 0.755 | 0.009 | ** | | | | |

According to the results, there are significant differences between the questions (in contrast to the *t*-tests per question). Looking at the model, we observe that Blackboard and Canvas users do not agree with finding course materials (Q2) or submitting assignments (Q3) through LMS. Therefore, they do not believe that the above LMSs can increase their academic performance (Q4). In addition, Blackboard and Canvas users do not agree with the understandability (Q8) and accessibility (Q11) of the LMSs. Additionally, Blackboard and Canvas users have no intention to check their grades through LMS (Q14) and to encourage their peers to use LMS (Q15). Furthermore, Blackboard and Canvas users are strongly uncertain about accepting LMS as a system that makes them happy (Q18). So, they cannot consider the LMSs as an innovative idea to download course materials (Q20/Q26), submit assignments (Q19/Q27), and check their grades (Q28). That is why they do not intend to tell their fellows about their satisfaction with using LMS (Q30).

*4.3. Descriptive Analysis (Construct Level)*

Table 6 and Figure 6 provide information similar to Table 3 and Figure 5. The mean, standard deviation, and Gini heterogeneity index values are provided per construct (instead of per question). Furthermore, the *t*-tests are applied to each construct to examine the statistical relationship between LMSs for each of the constructs. (Note that the values in Table 2 showed that the questions per construct provide consistent results, which allows us to combine these values and analyze constructs).

**Table 6.** Mean (*M*), standard deviation (*SD*), and Gini heterogeneity index (Gini) values for each TAM construct for Blackboard and Canvas LMSs. In addition, the *t*-values of the *t*-tests comparing the results per construct between the systems and the corresponding *p*-values are provided. At the bottom of the table, the mean and standard deviations over all questions are provided.

|  | Blackboard | | | Canvas | | | | |
|  | *M* | *SD* | Gini | *M* | *SD* | Gini | *t* | *p* |
|---|---|---|---|---|---|---|---|---|
| PU | 3.784 | (0.664) | 0.095 | 4.003 | (0.535) | 0.075 | 0.157 | 0.876 |
| PEU | 3.718 | (0.687) | 0.103 | 4.314 | (0.628) | 0.080 | 0.150 | 0.881 |
| BIT | 4.009 | (0.587) | 0.081 | 3.837 | (0.630) | 0.092 | 0.140 | 0.889 |
| ATUT | 3.627 | (0.666) | 0.104 | 3.724 | (0.636) | 0.094 | 0.215 | 0.830 |
| ATU | 3.960 | (0.565) | 0.079 | 3.953 | (0.638) | 0.090 | 0.176 | 0.861 |

The *t*-test results indicate that there are no significant differences between the systems for each construct ($p > 0.5$).

Looking at Figure 6, we see a relatively similar picture to Figure 5. Again, some Blackboard constructs are in the top left corner, indicating that the mean values for the constructs are lower than that of Canvas, but also the Gini heterogeneity index values are higher, indicating a less consistent answer selection by the Blackboard users.

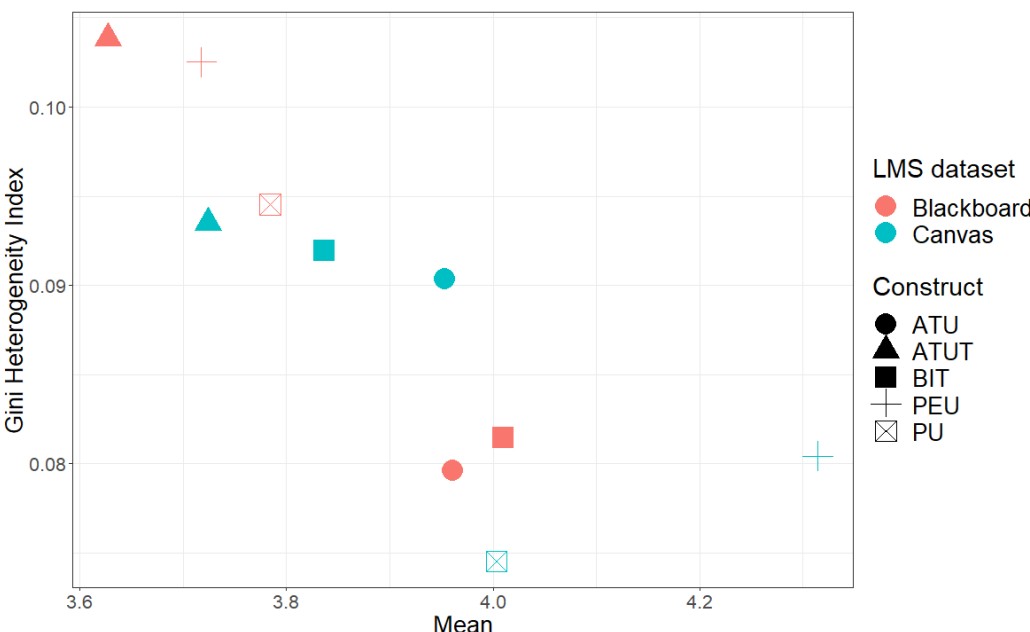

**Figure 6.** Descriptive summary from the answers to the TAM constructs and the variability between the answers per construct for Blackboard and Canvas.

Considering the distribution of the constructs, we see that for Blackboard and Canvas, the results for ATUT are the lowest. BIT is the highest value for Blackboard, but it is (after ATUT) the lowest for Canvas. This shows that the acceptance of the systems is different based on different properties.

*4.4. Cumulative Link Mixed Model (Construct Level)*

Similar to the analysis performed on a question basis, we here investigate the relationships between LMSs and the TAM constructs. Again, we build a Cumulative Link Mixed Model, which predicts the most likely answer. However, here, the answers are grouped per TAM construct, but we keep the LMS into account. Note that this can be accomplished as the questions related to the different TAM constructs show consistent behavior. This CLMM results in a linear model using the weights of the LMS and the TAM constructs.

Similar to the model that fits the answers based on the questions (and the LMS), here, we also consider possible interactions between the LMS and TAM constructs. Again, the participant variable is a random variable in the model. Table 7 provides all the weights and their $p$-values of the model. Blackboard and ATU are taken as the reference values.

**Table 7.** Weights and $p$-values for the two variables (LMS and TAM constructs) indicating the significant influences of the values for the variables in the Cumulative Link Mixed Model, where "***" indicates $p < 0.001$.

| Coefficients | Weight | $p$ | |
|---|---|---|---|
| Canvas | 0.031 | <0.001 | *** |
| PU | −0.205 | <0.001 | *** |
| PEU | 0.219 | <0.001 | *** |
| BIT | −0.111 | <0.001 | *** |
| ATUT | −0.741 | <0.001 | *** |
| Canvas-PU | −0.275 | <0.001 | *** |
| Canvas-PEU | −0.823 | <0.001 | *** |
| Canvas-BIT | 0.169 | <0.001 | *** |
| Cavas-ATUT | −0.169 | <0.001 | *** |

According to the model, there are significant differences between all constructs (in contrast to the $t$-tests per construct). Considering the ATU construct as a reference and the CLMM results for the questions, you can see a significant difference in particular between the PU and BIT constructs. This means that the Blackboard and Canvas users do not agree with the LMSs' usefulness (PU). That is why they do not intend to use the systems or encourage their peers to use the LMSs (BIT).

*4.5. Descriptive Network Analysis (Construct Level)*

4.5.1. Degree Centrality

According to the normalized coefficient of variation of the degree distribution depicted in Figure 7, there is a clear trend of low variability for "more than 3" networks compared with "equal to 3" and "less than 3" networks for both Blackboard and Canvas users. In other words, they tend to have a similar perception about the acceptance of the LMSs mostly for the aspects they answered with a high score. This is also due to the high rate of 4 and 5 answers, which increases the likelihood that users give the same answers to the same questions.

What is interesting for Blackboard is the low variability of almost all networks except for BIT and PU with variability of over 20% for the "less than 3" networks. Canvas, on the other hand, has higher variability on average for each structure and network type. This is especially noticeable for the PEU construct of the "less than 3" and the "equal to 3" networks. This means that the perception of the acceptance is more variable with the use of Canvas than with Blackboard; the users' low answers make its definition even more evident.

Therefore, in general, the low variability of degree centrality distribution (concerning 4 and 5 answers, which are the more frequent answers) indicates a certain consistency in users' answers: namely, both Blackboard and Canvas users have the same understanding of different features of the LMSs.

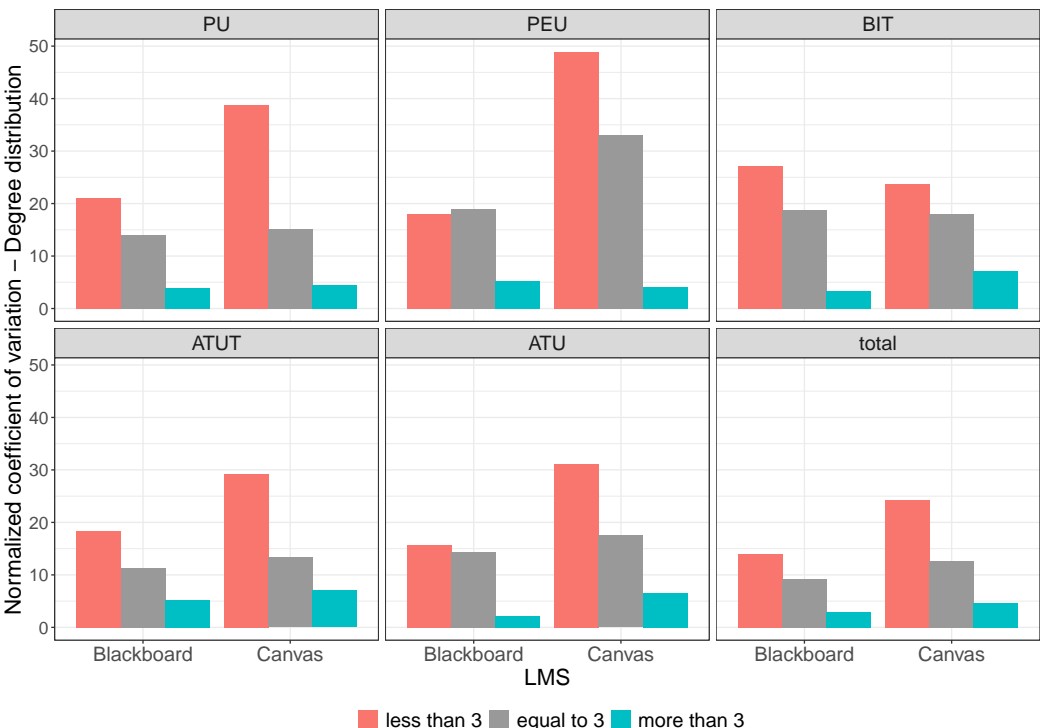

**Figure 7.** Normalized coefficient of variation of the degree centrality distribution for Blackboard and Canvas users.

4.5.2. Bipartite Motifs

As shown in Figures 8 and 9, the results prove the frequency of high answers for motifs for the two LMSs. As answers are related to the ratio of nodes, we can then compare answers on the basis of the network types. According to the details provided in the paragraphs covered in Section 3, the most intriguing extreme theme combinations are those with one or two users and a large number of questions (3, 4, or 5), or numerous users (3, 4, or 5) with a small number of questions (one or two). They outline patterns of questions that elicit the same answers from users or groups of users who answer similarly to the questions. This is consistent with the following motifs: 4, 7, 8, 12, 16, 17, 23, and 44 (see Figure 3).

Looking at the "less than 3" networks, we observe a global different behavior for both LMSs, specifically with Blackboard having higher relative frequencies of PU, PEU, and ATUT constructs than Canvas. This corresponds to the motifs 1, 3, 5, 7, 17, 38, 40, 42, and 44. This means that the above motifs are very important for PU, PEU, and ATUT compared to the other two TAM constructs. While the above TAM constructs (i.e., PU, PEU, and ATUT) for the first seven motifs with less than five nodes (Figure 8) have high frequencies, the graph shows lower frequencies for the remaining motifs (Figure 9). This means that low scores are fairly evenly distributed across both LMSs; however, the similarity between PU (Q1–6), PEU (Q7–11), and ATUT (Q16–23) is interesting with respect to the TAM (see Figure 1), and the ATUT construct derives directly from the PU and PEU constructs. With respect to the questions (Q1–6, Q7–11, and Q16–23), this means that both Blackboard and Canvas users do not concur that the above LMSs are helpful to their learning process. In addition, they do not consider the LMS as an innovative idea to submit their assignment or to download their course materials. In addition, the users do not accept the usability of the LMSs in terms of easy operation, understandability, and accessibility as well as the system interface. Therefore, they do not intend to use the system frequently or to encourage their peers to do so.

According to the "equal to 3" networks, for Blackboard, for the first seven motifs with less than five nodes (Figure 8), we observe the same behavior as "less than 3" networks

with the high proportion of neutral answers for PU and ATUT than other constructs in Blackboard. While for Canvas, you can see the higher peaks at motifs 5, 13, 17, 38, and 40 for BIT, ATU, and ATUT which are the patterns of users that have given a similar answer to a question or the patterns of questions that receive the similar answers by a user. In the meantime, for the remaining motifs (Figure 9), the two LMSs demonstrate the lower frequencies with a steady decline for Blackboard. With respect to the TAM (see Figure 1) and ATUT being derived directly from the PU construct, the resemblance between PU (Q1–6) and ATUT (Q16–23) constructs for Blackboard can be intriguing. This indicates that the Blackboard users do not care that the Blackboard LMS is beneficial to their learning process. Furthermore, they do not consider the Blackboard LMS to be a novel way to submit assignments or download course materials. Therefore, they might or might not encourage fellows to use the system. In the meantime, the resemblance between BIT, ATUT, and ATU constructs can be thought-provoking, since BIT derives directly from ATUT and also from ATU from BIT constructs. In light of the questions (Q12–15, Q16–23, Q24–30), this suggests that the Canvas users are neutral about using the system frequently or encouraging their peers to do so because it makes no difference to them that the Canvas LMS is a novel idea of submitting assignments or downloading course materials. That is why they might or might not be satisfied with the system.

Finally, for the "more than 3" network, what is striking is a sharp increase for all constructs and high frequencies for almost all motifs for both LMSs. This depicts the patterns of questions that receive similar answers from a user or patterns of users that give similar answers to a question for the above motifs. In addition, the overlap between ATU (Q24–30) and BIT (Q12–15) in Blackboard as well as the high frequency of PU (1–6) and PEU (Q7–11) in Canvas is very interesting. Considering the TAM (see Figure 1), ATU derives directly from the BIT construct and PU moves exactly in line with PEU. According to the questions in Blackboard (Q24–30, Q12–15), the Blackboard users are happy with the LMS and have the intention to use it regularly. Regarding the questions (Q16–23, Q7–11) in Canvas, this suggests that the Canvas users think choosing the LMS is a smart choice in terms of usability and accessibility. They believe that Canvas LMS is useful for increasing learning productivity as well as submitting assignments or downloading course materials. In the meantime, PEU in Canvas has higher frequencies than other constructs. This indicates that Canvas users have fully accepted the usability of the LMS compared to Blackboard users.

As a result, what is interesting once looking at the above figures is the similar behavior of Blackboard and Canvas for the "more than 3" network. This can be immediately observed in the "more than 3" networks: the motif answers are always higher than the "less than 3" and "equal to 3" networks for the two LMSs. This is the pattern of users that give a similar answer (high scores) to a question or the patterns of questions that receive similar answers (high scores) by a user. Furthermore, it seems that the most important constructs are ATU and BIT for Blackboard plus PU and PEU for Canvas. As mentioned above, with respect to the TAM, this makes sense. Since PU moves along the PEU construct and ATU extracts from the BIT construct, this means that the system's usability and ease of use interact in some way (for Canvas users). Meanwhile, users' intention to use the LMS can have an impact on their actual use of the system (for Blackboard users). Furthermore, with respect to the questions, the Blackboard users are delighted with the system and want to use the Blackboard LMS frequently. Likewise, Canvas users see LMS as a sensible and reasonable choice with regard to usefulness and usability. They believe it would be wise to discuss this with the instructor and peers. Nevertheless, both Blackboard and Canvas users do not consider using the LMS as an innovative idea and a wise decision. In relation to the motif, this means that for both Blackboard and Canvas users, the "more than 3" answers are higher than the "less than 3" and "equal to 3" answers. While the most important constructs for Blackboard users are their intention to use LMS and its actual use, the most essential constructs for Canvas users are their perception of the LMS usability and the system's ease of use.

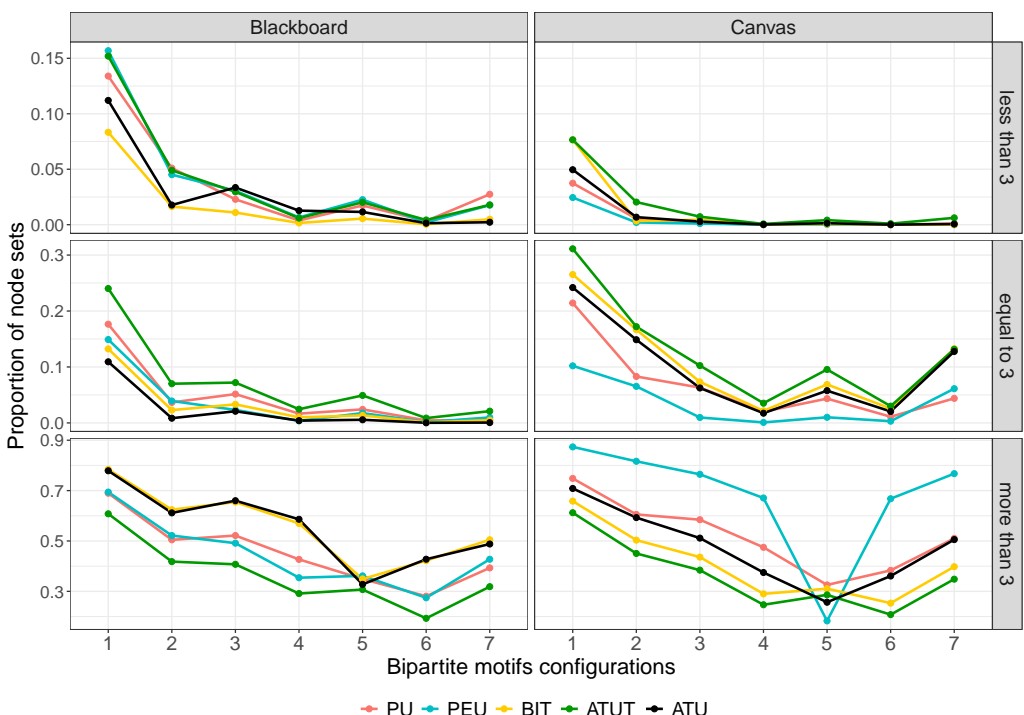

**Figure 8.** Relative motif frequencies on user/construct for "less than 5 nodes" configurations (Blackboard and Canvas).

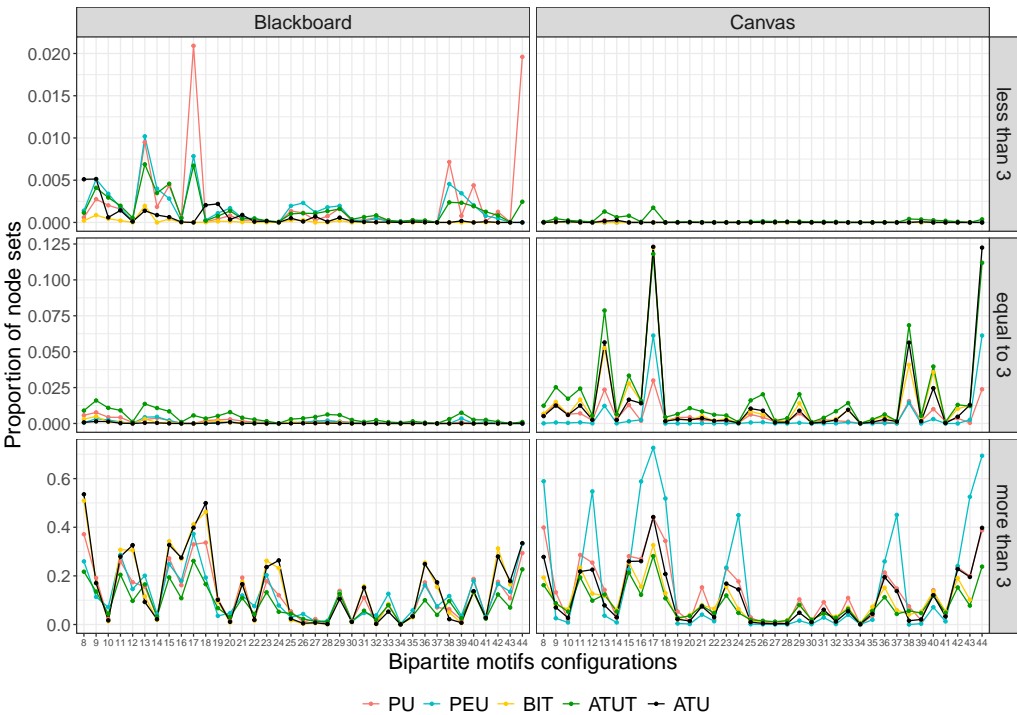

**Figure 9.** Relative motif frequencies on user/construct for "more than 4 nodes" configurations (Blackboard and Canvas).

Although the frequencies of the motifs configurations depend on the number of questions contained in a construct, we can conclude that the use of motifs highlights the following: while for the "more than 3" networks, both Blackboard and Canvas show a global similar behavior (similar variations in the observed motifs configurations), for the "less than 3" and "equal to 3" networks, their behavior is reversed (evident from Figure 9).

## 5. Discussion

The aim of this article is to demonstrate the data science techniques and approaches for data collection, processing, evaluating, and analyzing the users' acceptance of the two LMSs: Blackboard and Canvas. We do this on two levels (both individual questions as well as TAM constructs) and through using different techniques. First, we applied statistical analysis, i.e., *t*-tests, to investigate whether there are significant differences between the two LMSs (or not). Second, we provide results utilizing descriptive indices, e.g., the mean, the standard deviation, and the Gini heterogeneity index to assess the general level of users' acceptance and the variability of their answers. Third, we estimated the effect of the questions and constructs as well as their interactions with the above LMSs on the users' answers through the Cumulative Link Mixed Model. Fourth, we experimented with descriptive network analysis, e.g., degree centrality and bipartite motifs to see the variation of users' answers and the patterns of users' satisfaction within TAM constructs.

On the question level, for the *t*-tests, we see that no significant differences can be found between the two LMSs. Similar results are found when analyzing the differences on the TAM construct level as well. There may be two reasons for this: Firstly, Blackboard and Canvas are both developed in the US with approximately similar capabilities. Secondly, the user groups are relatively similar, namely both systems were evaluated in the [2], mostly by users who studied at the School of Humanities and Digital Sciences with somewhat similar experiences of LMSs. Regarding the consistency of answering the questions, we observe that overall, the participants are very consistent (when looking at the low Gini values). Additionally, the standard deviations are relatively small, indicating that not only did participants typically select the same answers, but the answers they select are also close together. For instance, if participants select mostly either answer 1 or answer 5, this would lead to a relatively low Gini score but a relatively high standard deviation. In the data, both Gini and standard deviation values are low.

When comparing consistency between LMSs, we see that Blackboard has slightly larger standard deviations and Gini scores than Canvas. For Blackboard, participants are somewhat less consistent in answering their questions compared to Canvas. While considering the results from the CLMM, we see that Blackboard and Canvas behave somewhat differently. There are some differences between the questions, but it is difficult to find clear patterns. Some questions are functionality specific (such as Q2 and Q3, which deal with finding course materials and submitting assignments). However, most questions of the PU and BIT TAM constructs show significant differences (both on the question and construct level, with Q1 and the ATU construct as a reference). In addition, the weight is negative for both constructs, but it is slightly lower for the PU.

For the network analysis, we examined bipartite motifs and degree centrality based on TAM constructs in the unweighted bipartite networks and weighted unipartite networks, respectively, which are derived from the three networks: "less than 3", "equal to 3", and "more than 3". As estimated, according to the motifs, network "more than 3" presented a rising trend and higher consistency compared to the other two networks. What stands out was the high frequencies of ATU and BIT for Blackboard as well as PE and PEU for Canvas. This means that most users gave high answers to the above constructs with respect to the other TAM constructs. Meanwhile, according to the TAM (see Figure 1), PU moves exactly along the PEU construct and ATU derives directly from the BIT construct. This striking resemblance between the pattern of motifs and TAM structure is very interesting. This proves how the LMS usability and the system's ease of use are related. At the same time, the actual use of the LMS by users originates from their intention to use the system. For the degree centrality distribution, the surprising result was the higher variability of BIT and PU for Blackboard (the same way as CLMM) as well as the PEU for Canvas. This striking similarity between the BIT and PU constructs is interesting. Looking at the TAM model, the BIT directly drives from the PU construct, meaning that Blackboard users do not accept the usability of the LMS. That is why they do not intend to use the system frequently or encourage their colleagues to do so. Furthermore, the remarkable variability of the PEU

construct for Canvas is intriguing. This indicates that Canvas users have a more variable perception of the LMS ease of use than Blackboard users. These results seem to suggest that most Canvas users find the system easy to use, but they are not necessarily fully satisfied with the system in its actual usage.

Overall, this study extends the previous work [2] (where only Blackboard was analyzed) to a larger context (Blackboard plus Canvas) focusing more on extra methodologies. In the previous study [2], basic statistical analysis, as well as network analysis, was applied to investigate the acceptance of the Blackboard LMS. Descriptive network analysis showed the consistency of users' perspectives toward the system; then, the descriptive statistics results enabled the extraction of actions related to the Blackboard LMS. In the current study, we provided a broader comparison and in-depth comprehension of user acceptance of the two systems (Blackboard versus Canvas). Therefore, we experimented with the general level of acceptance and its heterogeneity plus the Cumulative Link Mixed Model (CLMM) at the question and construct level through descriptive analysis as well as network analysis at the construct level. The results confirmed the higher acceptance and consistency among Canvas users compared to Blackboard, which helped the Tilburg University LMS group while switching LMSs from Blackboard to Canvas.

In comparison to the previous study [2], the strengths of the current study were employing a Cumulative Link Mixed Model to describe user acceptance by estimating the probability of the users' answers with respect to the different LMSs and the questions/constructs. Additionally, the network analysis approaches used in the current study revealed interesting patterns in the participant data to more precisely describe the user acceptance of LMSs. Nevertheless, the generalizability of these results is subject to certain limitations. For instance, no major strategic conclusions can be made based on the results from this study, as these results are based on a small sample from one university in The Netherlands. Another limitation of this study is that demographic information has not been taken into account. Previous work [2] has shown that gender may have an impact on the results (although that study did not show a considerable influence).

## 6. Conclusions

In this article, we presented several analyses of questionnaire results that investigate the acceptance of two learning management systems (LMSs), namely Blackboard and Canvas. The analyses were performed on two levels: questions and constructs that stem from the Technology Acceptance Model (TAM), which also formed the basis for the questions in the questionnaire.

We compared the LMSs using statistical properties. We provided descriptive measures, e.g., mean, standard deviations, and Gini heterogeneity index results for both levels. The means provide overall values for the answers to the questions in the questionnaire, whereas the standard deviation shows the spread. The Gini heterogeneity index indicates the consistency in the answers. We also applied the Cumulative Link Mixed Model (CLMM) to both levels. The Cumulative Link Mixed Model examines the impact of the questions and TAM constructs with the LMSs on the users' answers. Finally, we experimented with descriptive network analysis, e.g., degree centrality and bipartite motifs to see the variability of users' answers and extract the patterns of user satisfaction across TAM constructs.

The results showed that overall, participants were very consistent in providing their answers. Both standard deviations and Gini heterogeneity scores were low for both questions and TAM constructs. The overall scores were high, indicating that participants seem to accept the use of LMSs. The Blackboard system, however, showed slightly lower scores compared to the Canvas system. We propose that this may be due to the differences in functionality of the two LMSs: namely, Canvas is somewhat better and more innovative in design than Blackboard.

Investigating the combination of metrics provides a fine-grained analysis of the results. Not only a statistical model is built which shows differences: the model is applied to two levels, which illustrates the differences in the construct as well as individual questions. The

use of the Gini heterogeneity index provides additional information on the consistency of answers between the participants (which may be different from the spread measured by the standard deviation). Finally, through descriptive network analysis, for both Blackboard and Canvas, we observed high equilibrium, which was due to a large proportion of satisfaction among users. For the Canvas users, however, the perception of the LMS acceptance was higher than the Blackboard users per construct.

As mentioned above, the reason for choosing the two LMSs in our data analysis of user acceptance is due to the academic context. The context for both LMS groups was almost the same: namely, the same university, the same educational program, and the same environment (both LMS users studied at the School of Humanities and Digital Sciences, and the LMSs were evaluated in The Netherlands).

Overall, the main goal of the current study was to illustrate how data science methodologies were applied for data collection, processing, assessment, and analysis in a particular context, namely the LMSs: Blackboard and Canvas. The empirical findings contribute in several ways to a new understanding of the LMSs and provide a basis for the analysis of LMSs' user acceptance. Firstly, the *t*-tests indicate that there are no significant differences between the two LMSs. Secondly, what is interesting is the relatively higher standard deviations and Gini scores for Blackboard than Canvas, meaning that Blackboard users are less consistent in answering their questions. Thirdly, looking at the CLMM results, you can see that Blackboard and Canvas users behave differently in answering some functional questions and constructs, i.e., Q2 and Q3 (finding course materials and submitting assignments) as well as PU and BIT (LMSs usefulness and users' intention to use the LMSs). Lastly, according to network analysis, namely bipartite motifs, the high frequencies of ATU and BIT for Blackboard, as well as PE and PEU for Canvas, are particularly noticeable. This indicates that compared to the other TAM constructs, the above constructs received high answers from the majority of users. Additionally, this proves the relationship between LMSs' usability and ease of use. Furthermore, the surprising result for the degree centrality distribution is the higher variability of BIT and PU for Blackboard as well as PEU for Canvas. This indicates that Blackboard users have no intention of using the system frequently or encouraging their colleagues to do so. The considerable variability of the PEU construct for Canvas, however, is unexpected, meaning that Canvas users have a more varied perception of the LMS's usability than Blackboard users. These findings appear to suggest that while the majority of Canvas users regard the system as easy to use, they are not always completely satisfied with it in practice or in actual use.

For future work, we would like to further investigate the underlying reasons for the differences between the acceptance of the LMS systems. This can be completed by separately querying students from different educational backgrounds and comparing these results. Additionally, the cultural (geographic) differences can be investigated further as well.

**Author Contributions:** The authors' contributions are as follows: conceptualization: P.S., R.R., M.v.Z. and M.A.; methodology: P.S., R.R., M.v.Z. and M.A.; software: P.S. and R.R.; validation, P.S., R.R., M.v.Z. and M.A.; formal analysis: P.S., R.R., M.v.Z. and M.A.; investigation: P.S. and R.R.; resources: P.S.; data curation: P.S.; writing—original draft preparation: P.S. and R.R.; writing—review and editing: P.S., R.R., M.v.Z. and M.A.; visualization: P.S. and R.R.; supervision: M.v.Z. and M.A.; project administration: M.A.; funding acquisition: M.A. All authors have read and agreed to the published version of the manuscript.

**Funding:** This research was funded by the German Research Foundation (DFG) project "MODUS" grant number AT 88/4-1.

**Institutional Review Board Statement:** The study was conducted in accordance with the Declaration of Tilburg, The Netherlands, and approved by the Ethics Committee of the School of Humanities and Digital Sciences School at Tilburg University (protocol code REC # 2018/60 and date of approval 18 January 2019).

**Informed Consent Statement:** Informed consent was obtained from all subjects involved in the study.

**Data Availability Statement:** Data supporting reported results can be found in the link below: https://github.com/p877/MDPI (accessed on 5 December 2022).

**Acknowledgments:** The research leading to this work has partially been funded by the German Research Foundation (DFG) project "MODUS" under grant AT 88/4-1.

**Conflicts of Interest:** The authors declare no conflict of interest. The funders had no role in the design of the study; in the collection, analyses, or interpretation of data; in the writing of the manuscript; or in the decision to publish the results.

## Abbreviations

The following abbreviations are used in this manuscript:

| | |
|---|---|
| LMSs | Learning Management Systems |
| TAM | Technology Acceptance Model |
| CLMM | Cumulative Link Mixed Model |
| EV | External Variables |
| PU | Perceived Usefulness |
| PEU | Perceived Ease of Use |
| ATUT | Attitude Toward Using the Technology |
| BIT | Behavioral Intention to use the Technology |
| ATU | Actual Technology Use |
| UTAUT | Unified Theory of Acceptance and Use of Technology |
| TPB | Theory of Planned Behavior |
| OSAM | Online Shopping Acceptance Model |
| PCA | Principal Component Analysis |
| AVE | Average Variance Extracted |
| CR | Composite Reliability |
| SD | Standard Deviation |
| M | Mean |

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
