# Peer review of "Multi-Level Analysis of Learning Management Systems’ User Acceptance Exemplified in Two System Case Studies†"

_data, 2022_

Round 1

Reviewer 1 Report

This article consists of carrying out a Multi-Level Analysis of Learning Management Systems’ User Acceptance Exemplified in Two Real-World Case Studies, especially you analyze data on user acceptance for two LMSs (Blackboard and Canvas) for multi-level analysis by using a questionnaire modeled after the Technology Acceptance Model (TAM) and Cumulative Link Mixed Model (CLMM)…, this subject which is topical and interesting.

The article is well written and the research topic is relevant but it should clarify the following points:

1-  Overall, the study shows good results. However, there are a lot of presentation issues in the paper. You must describe what are the reasons for choosing the two LMSs in your data analysis of user acceptance?

2-  you must mention citation numbers [?].

3-  Add more relevant references.

4-  Extend more the Result and related work

Reviewer 2 Report

Thank you for the opportunity to review this article. I attached my recommendations. 

Reviewer 3 Report

Title does not accurately reflect the nature of the paper. The paper demonstrates some statistical techniques using the example of questionnaires applied on users of two LMS software namely Canvas and BB. A more appropriate title is required that will clearly guide the readers. 

ln167 what is meant by basic fucntionality

ln110 rephrase "nicely fits"

ln 251 what is a "regular" statistical method?

ln 265 Likert-style you have already used and repeated many times "Likert scale" which is always used to explain a certain question type but Likert style seems vague

ln324 delete "(gender and age)" obviously demographic explains it

Table 3 is completely redundant

ln383 why do you use three decimal points the numbers are almost identical in addition I have never seen the average reported for all responses over all questions

ln488 delete remember

ln699 important note the aim of the paper cannot be making any strategic conclusions based on such a small sample from a university in the Netherlands rather as per the note on data's web site the aim is to demonstrate the collection, treatment and analysis methods of data in science. Your aim therefore is to demonstrate some novel methods in education science. The title (as noted earlier) ad other parts should be modified accordingly. Also make sure to use a more formal language.
